# Design and Implementation of Ternary Logic Integrated Circuits by Using Novel Two-Dimensional Materials

**Mingqiang Huang [1], Xingli Wang [1], Guangchao Zhao [1], Philippe Coquet [1,2] and Bengkang Tay [1,*]**

[1] CNRS-International-NTU-THALES-Research-Alliance (CINTRA), Nanyang Technological University, Singapore 639798, Singapore; mqhuang@ntu.edu.sg (M.H.); wangxingli@ntu.edu.sg (X.W.); ZHAO0303@e.ntu.edu.sg (G.Z.); PCoquet@ntu.edu.sg (P.C.)

[2] Institut d'Electronique, de Microélectronique et de Nanotechnologie (IEMN), CNRS UMR 8520-Université de Lille, 59650 Villeneuve d'Ascq, France

[*] Correspondence: EBKTAY@ntu.edu.sg; Tel.: +65-6790-4533

**Abstract:** With the approaching end of Moore's Law (that the number of transistors in a dense integrated circuit doubles every two years), the logic data density in modern binary digital integrated circuits can hardly be further improved due to the physical limitation. In this aspect, ternary logic (0, 1, 2) is a promising substitute to binary (0, 1) because of its higher number of logic states. In this work, we carry out a systematical study on the emerging two-dimensional (2D) materials ($MoS_2$ and Black Phosphorus)-based ternary logic from individual ternary logic devices to large scale ternary integrated circuits. Various ternary logic devices, including the standard ternary inverter (STI), negative ternary inverter (NTI), positive ternary inverter (PTI) and especially the ternary decrement cycling inverter (DCI), have been successfully implemented using the 2D materials. Then, by taking advantage of the optimized ternary adder algorithm and the novel ternary cycling inverter, we design a novel ternary ripple-carry adder with great circuitry simplicity. Our design shows about a 50% reduction in the required number of transistors compared to the existing ternary technology. This work paves a new way for the ternary integrated circuits design, and shows potential to fulfill higher logic data density and a smaller chip area in the future.

**Keywords:** 2D materials; black phosphorus; inverter; ternary logic; adder

## 1. Introduction

Integrated circuits (IC) are the cornerstone of modern information society and are widely used in almost all of the electronic systems. Binary digital integrated circuits are the most prevalent computing technology today, because the binary signals exist naturally in semiconductor electronic devices, and binary Boolean Algebra can be easily implemented by using binary logic gates. However, the performance of binary logic is fundamentally limited by its low density of logic states, that is, only two logic levels (0, 1) can be transmitted over a given set of lines [1]. Therefore, it needs a large number of logic gates and transistors to reach the required data size. Besides, even more interconnected wirings between the system components are required in integrated circuits. For example, in a very large scale integrated (VLSI) circuit, approximately 70 percent of the area is devoted to interconnection, 20 percent to insulation, and only 10 percent to device [1–3], which exceedingly increases the complexity, both in design and manufacture.

A ternary digit, which is also called as a trit, owns three significant values (0, 1, 2), and can represent $\log_2(3) = 1.58$ bits [4–6]. Thus, more information can be transmitted over the interconnections,

and less devices are required for a given data length. For instance, a ternary system with a digit size = 19 can represent a data size about $3^{19} \sim 1$ G. Meanwhile, that of a binary system is only $2^{19} \sim 0.5$ M.

Ternary logic has been widely studied for decades, but is still severely blocked by the fact that the designs of ternary are much more complex than those of binary. In the 1980s, the design of silicon CMOS ternary occurred as a moderate breakthrough, where one p-MOSFET, two resistors and one n-MOSFET were in-series connected [4,5]. Such circuits simplified the design, but it required two passive resistive resistors, and increased the production complexity. More seriously, it largely increased the static power consumption because of the two resistors with relative low resistance. In the 1990s, the carbon nanotube field effect transistor (CNTFET) showed itself to be promising for its ballistic transport, high mobility and low off-current properties [7,8]. And it was then found that the threshold voltage of CNTFET could be well determined by its diameter, which was quite suitable for the ternary threshold logic design [9,10]. Based upon that, many groups have focused their attention on the demonstration of ternary logic gates using multi-threshold CNTFETs [11–15]. In their designs, no passive resistive components were needed, but the simplest ternary NOT gate still needed six CNTFETs, which was much more complicated than that of a classical CMOS binary NOT gate (two FETs). Therefore, both the silicon ternary and CNT ternary are not cost-efficient enough to establish a practical ternary application: It needs four transistors in silicon ternary and six transistors in CNT ternary to process one trit (which equals 1.58 bits) of information. Meanwhile, one can use the same four (or six) transistors to process two (or three) bits of information in binary logic circuits. A recent breakthrough on ternary logic is the demonstration of a ternary NOT gate by using the commercial CMOS processes [16]. But it suffers from the low operation frequency due to the ultra-low current, and the other types of ternary logic functions are still undiscovered.

Recently, the emerging two-dimensional (2D) semiconducting materials, such as graphene, transitional metal dichalcogenides (TMDs) and black phosphorus (BP), have attracted enormous attention due to their excellent electronic properties [17–23], and this also opens up new possibilities in logic circuits in terms of both CMOS binary [24,25] and ternary logic design [26–31]. For example, Tomas Palacios et al. [26] presented the first 2D-materials-based standard ternary inverter in 2016. Huang et al. [28] designed and fabricated the novel tunable $MoS_2$/BP-based ternary devices (tuned by the electric field and device channel length). Jin-Hong Park et al. [29] demonstrated the graphene/$WSe_2$-based light triggered ternary device. However, all of these previous works are mainly focused on the demonstration of a standard ternary inverter (STI). To accomplish simplicity in circuit design and increase the data density in practical integrated circuits, further research on ternary logic devices, circuits and algorithms needs to be explored. In this work, we propose to perform a systematical study on the 2D-based ternary logic from individual ternary logic gates to large scale ternary integrated circuits. We will firstly utilize the unique electronic properties of ambipolar BP transistors and N-type $MoS_2$ transistors to build various ternary logic gates. Then we will integrate the as-demonstrated logic gates to design and realize the dyadic ternary operators such as T-NAND, T-NOR and other ternary functions. Finally, we will focus on the design of large scale ternary integrated circuits applications. The decrement cycling inverter and the optimized ternary adder algorithm will be used to design the 19-trit ternary adder.

## 2. Demonstration of Ternary Logic Gates

In conventional binary Boolean Logic, there are a total of $2^2 = 4$ monadic functions. While in ternary logic [6], the number of monadic functions is $3^3 = 27$. These 27 ternary functions are enumerated from 0 to 9 and then A to Z, as shown in Figure 1a. Strictly speaking, we do not need all of these functions because there are strong relationships between such operations in Boolean Algebra. Several of them are nontrivial and meaningful [6]. For example, functions 0, D, and Z are the trivial, constant-valued functions; Function P is identity; Function 5 is the standard ternary inverter (STI); Functions 2 and 8 are the negative threshold inverter (NTI) and positive threshold inverter (PTI), respectively; Functions 7 and B are increment cycling inverter (ICI) and decrement cycling inverter

(DCI), respectively. It is widely accepted that the most important and fundamental components in ternary logic are STI, NTI and PTI [9–12]. The output-input characteristic curves of such ternary functions have been shown in Figure 1b.

STI is actually a ternary NOT function that inverts its input, returning "0" when given "2" and vice versa, while leaving "1" inputs unchanged; NTI inverts the input of "2" and "0" as same as the NOT function, but returns "0" when given "0"; PTI inverts the input of "2" and "0", but returns "2" when given "1". Cycling ternary (CT) gates can operate the increment function (output = input + 1) or decrement function (output = input - 1), which are especially suitable for the ternary ripple-carry adder. While we still do not need both of them, since (a − 1) = ((a + 1) + 1) = −(−a + 1), which means a decrement gate can be achieved by two increment gates, or by two NOT gates together with one increment gate. Similarly for the increment gate, it can be achieved by using two decrement gates, or by two NOT gates plus one decrement gate because (a + 1) = ((a − 1) − 1) = −(−a − 1).

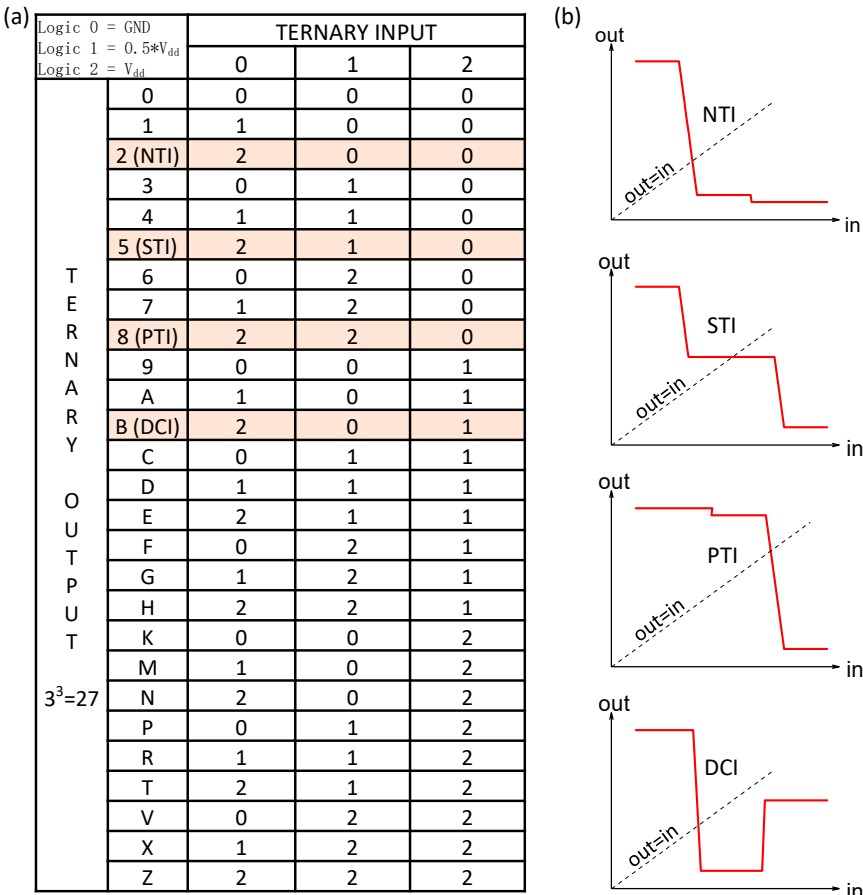

(a)

Logic 0 = GND
Logic 1 = 0.5*Vdd
Logic 2 = Vdd

TERNARY OUTPUT　3³=27

| | TERNARY INPUT | | |
|---|---|---|---|
| | 0 | 1 | 2 |
| 0 | 0 | 0 | 0 |
| 1 | 1 | 0 | 0 |
| 2 (NTI) | 2 | 0 | 0 |
| 3 | 0 | 1 | 0 |
| 4 | 1 | 1 | 0 |
| 5 (STI) | 2 | 1 | 0 |
| 6 | 0 | 2 | 0 |
| 7 | 1 | 2 | 0 |
| 8 (PTI) | 2 | 2 | 0 |
| 9 | 0 | 0 | 1 |
| A | 1 | 0 | 1 |
| B (DCI) | 2 | 0 | 1 |
| C | 0 | 1 | 1 |
| D | 1 | 1 | 1 |
| E | 2 | 1 | 1 |
| F | 0 | 2 | 1 |
| G | 1 | 2 | 1 |
| H | 2 | 2 | 1 |
| K | 0 | 0 | 2 |
| M | 1 | 0 | 2 |
| N | 2 | 0 | 2 |
| P | 0 | 1 | 2 |
| R | 1 | 1 | 2 |
| T | 2 | 1 | 2 |
| V | 0 | 2 | 2 |
| X | 1 | 2 | 2 |
| Z | 2 | 2 | 2 |

(b) NTI, STI, PTI, DCI characteristic curves (out vs in, out=in reference line)

**Figure 1.** (**a**) List of the 27 ternary operators. (**b**) Characteristic curves of ideal negative ternary inverter (NTI), standard ternary inverter (STI), positive ternary inverter (PTI) and decrement cycling inverter (DCI) from top to down.

By utilizing the unique electronic properties of ambipolar BP transistors and N-type $MoS_2$ transistors, the above-mentioned ternary logic gates have been successfully designed and fabricated in this work. The specific fabrication process is described as following. Firstly, commercial-available few-layer black phosphorus and $MoS_2$ flakes were mechanically exfoliated onto a silicon substrate with a 20 nm $HfO_2$ high-k dielectric layer. Then the flakes were identified under a microscope in a glove box, where the oxygen and humidity were always kept below 0.1 ppm. Next, standard electron-beam lithography and electron-beam evaporation of 20 nm/60 nm Ni/Au were used to form the device structures and metal contacts. After that, the current-voltage performance of the devices was measured

in a probe station under high vacuum. BP FETs and MoS$_2$ FETs with a suitable threshold voltage and on-off ratio, as well as a high mobility, were chosen for the second electron-beam lithography and electron-beam evaporation to fabricate the interconnections.

Because the BP flakes were easily deteriorated in air conditions, great attention should be taken to protect the samples during the whole fabrication. The samples should always be kept in the glove box, and the time of exposure to air in-between process steps should be minimized.

For the BP-MoS$_2$-based standard ternary inverter (Figure 2a,b), unlike previous studies of a CMOS inverter based on a MoS$_2$ n-type MOSFET and a BP p-type MOSFET [24,25], the BP channel here is divided into two regions with a short channel of 0.1 μm and a long channel of 1.8 μm. We may simply name the in-series MoS$_2$ FET plus BP1 FET as Part 1, and the BP2 FET as the Part 2, respectively. Note that in our circuit design, the output voltage is always represented as $V_{out} = V_{dd}*R_1/(R_1 + R_2)$, where $R_1$ and $R_2$ is the resistance of Part 1 and Part 2, respectively. If the ratio of R1/R2 is constant, then the output voltage will also be a constant. Therefore, it will show a middle logic output ("1") when $V_{in}$ is middle. In addition, it is easily to get the other two states, namely "highest logic level" ("2") and "lowest logic level" ("0"). Thus, the output voltage will show three stable states. The NTI and PTI differ from STI only when the input is middle. Note that $R_1$ and $R_2$ are mainly controlled by the BP1 and BP2, and the MoS$_2$ resistance is almost negligible when input is middle [19,20]. Therefore, we can tune the middle output logic level to be very high for the PTI (Figure 2c) or very low for the NTI (Figure 2d) in our design.

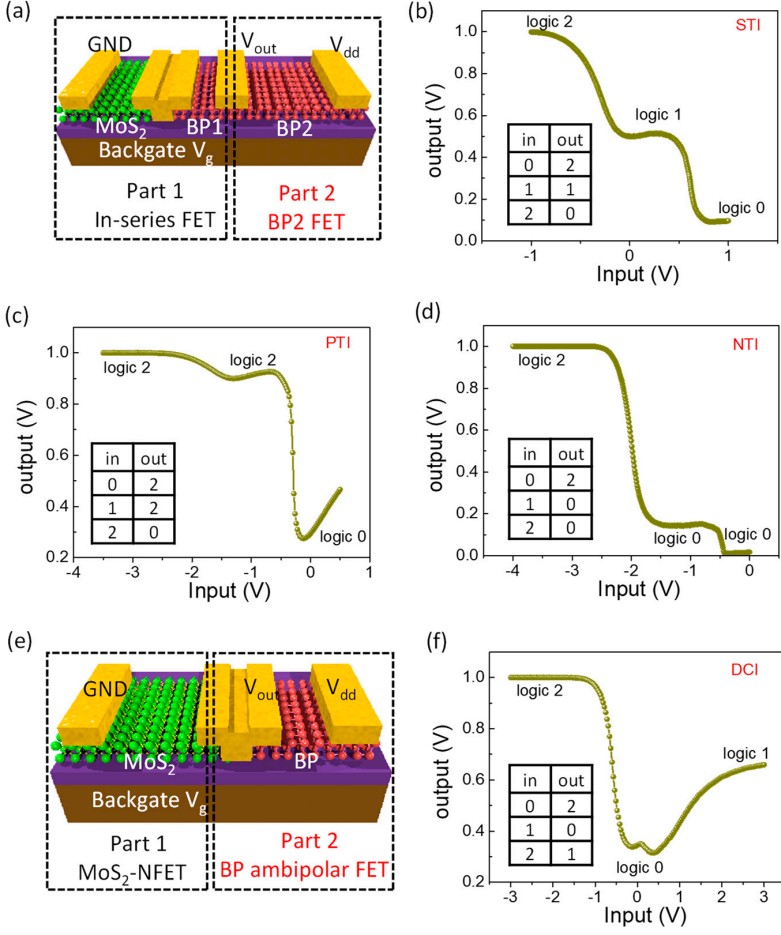

**Figure 2.** Demonstration of different ternary logic gates. (**a**). Schematic view of a BP-MoS$_2$ standard ternary inverter. $V_{out}$-$V_{in}$ characteristic curves of the (**b**) STI, (**c**) PTI and (**d**) NTI. (**e**). Schematic view of a BP-MoS$_2$ decrement cycling ternary inverter. (**f**). $V_{out}$-$V_{in}$ characteristic curves of the DCI.

Furthermore, beneficial from the ambipolar behavior of a BP transistor with high mobility of both hole and electron, we design and fabricate a decrement cycling inverter (DCI) with a quite simple device structure (Figure 2e,f). There is only one BP transistor, together with one N-type $MoS_2$ transistor used for the design of the DCI, which is just the same as a CMOS binary inverter [24,25]. When $V_{in}$ is low, $MoS_2$ FET is off while BP FET is on, then the output is high (logic "2"); when $V_{in}$ is middle, $MoS_2$ FET is on while BP FET is off, then output is low (logic "0"); when $V_{in}$ is further higher, both $MoS_2$ FET and BP FETs are fully turned on as the n-type transistor [23], then there will be a voltage distribution in between the resistance of $MoS_2$ and BP. We will get a middle output if the $R_{MoS2} = R_{BP}$ on its n-type which forms the third state of the inverter (logic "1").

So far, we have designed and fabricated four types of inverters (STI/NTI/PTI/DCI) by using 2D materials. Compared to the existing silicon ternary [4,5] and CNT ternary [9–15], our design has largely simplified the ternary logic design complexity. There are only two or three FETs involved in one ternary logic gate, which is only about half of that in silicon ternary or CNT ternary. Therefore, it is possible to accomplish simplicity in modern digital design by using ternary logic due to the reduced circuit overhead and chip area.

## 3. Design of Ternary Logic Circuits

### 3.1. Functional Ternary Logic Gates

The truth table of ternary NOT-AND (NAND) and ternary NOT-OR (NOR) gates are given in Figure 3a. The two-input ternary logic functions of T-NAND and T-NOR gates [11] are defined by the following two equations:

$$Y_{NAND} = \overline{min\{A, B\}} \tag{1}$$

$$Y_{NOR} = \overline{max\{A, B\}} \tag{2}$$

For the T-NAND, it contains two steps of logic operations: Firstly it minimizes the two inputs of A and B, and then outputs the result after a NOT transformation. If the minimum input is "0", the output is "2". If the minimum input is "1", the output is "1". If the two inputs are all "2", the output is "0". The designed circuits are shown in Figure 3b, where 4 BP transistors and 2 $MoS_2$ transistors are involved. According to our pre-demonstrated STI as shown in section II, the parameters of the transistors are designed to be $L_{ch} = 0.1$ μm for BP1 and BP1', $L_{ch} = 1.8$ μm for BP2 and BP2' and $L_{ch} = 1$ μm for the two $MoS_2$ FETs. The core question of T-NAND simulation will be the calculation of the resistance of each part at different input voltage. Here, one important thing which needs to be pointed out is that the effective source voltage $V_s$ of the pull-up part will be $V_{out}$, while that of the pull-down part is always $V_s = 0$. When input of A is "0", the $MoS_{2'}$ FET is fully turned off, so the output is "2", no matter what B is; When the input of A is "1", the BP1' FET plus the $MoS_{2'}$ FET is half turned on, so the output will not be "0" (but it will be "1" or "2", dependent upon B); When the input of A is "2", the BP1' FET plus the $MoS_{2'}$ FET is turned on as n-type, so the output is dependent on the input of B (Figure 3c).

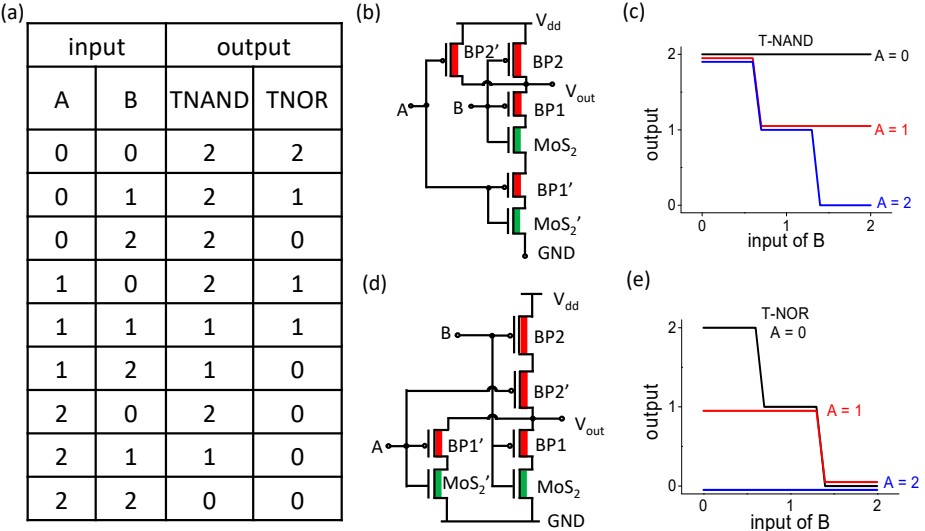

**Figure 3.** (**a**) The truth table of 2-input ternary NOT-AND (NAND) and NOT-OR (NOR) gates. The designed circuits (**b**) and $V_{out}$-$V_{in}$ characteristic curves (**c**) of the T-NAND gate. The designed circuits (**d**) and $V_{out}$-$V_{in}$ characteristic curves (**e**) of the T-NOR gate.

For the T-NOR, it also contains two-step of logic operations: It maximizes the two inputs of A and B, and then outputs the result after a NOT transformation. If the maximum input is "2", the output is "0". If the maximum input is "1", the output is "1". If the two inputs are both "0", the output is "2". The designed circuits for the T-NOR are shown in Figure 3d, where four BP transistors and two MoS$_2$ transistors are involved. The transistors' parameters are similar to that of a T-NAND. When the input of A is "2", the BP2′ FET is fully turned off, so the output is "0", no matter what B is; When the input of A is "1", the BP2′ FET is half turned on, so the output will not be "2" (yet it will be "0" or "1", dependent on B); When the input of A is "0", the BP2′ FET is turned on as a p-type, so the output is dependent on B (Figure 3e).

Ternary AND and ternary OR gates are known as the minimum and maximum gates. These two gates can be generated by adding one ternary NOT gate after T-NAND and T-NOR, respectively. Therefore, the two-input T-AND gate needs nine FETs (six for T-NAND and three for NOT), and the T-OR gate also needs nine FETs. Similar design methods can be used to implement the multi-input T-AND and T-OR logic gates.

The ternary logic gates presented in Figure 4a,b can be used for designing ternary arithmetic circuits. As required for the ternary circuits, a design of a ternary decoder [9–11] is presented in Figure 4a. The ternary decoder is a one-input, three-output combinational circuit, and generates unary functions for an input $X_k$. The response of the ternary decoder to the input $X$ is given by

$$X_k = \begin{cases} 2 & if \quad X = k \\ 0 & if \quad X \neq k \end{cases} \tag{3}$$

where $k$ can take logic values of "0", "1" or "2". Note that the outputs of the decoder have only two logic values (namely "2" and "0"), corresponding to highest logic state ("2") and lowest logic state ("0") in binary logic, therefore, binary OR gates can be used here. A ternary buffer is also presented in Figure 4a. The ternary buffer contains one DCI and one STI. It is easy to deduce that the response of the buffer is given by

$$X_{out} = \begin{cases} 1 & if \quad X_{in} = 2 \\ 0 & if \quad X_{in} = 0 \end{cases} \tag{4}$$

(a)　　　　　　　　　　　　　　　　　　　　　　(b)

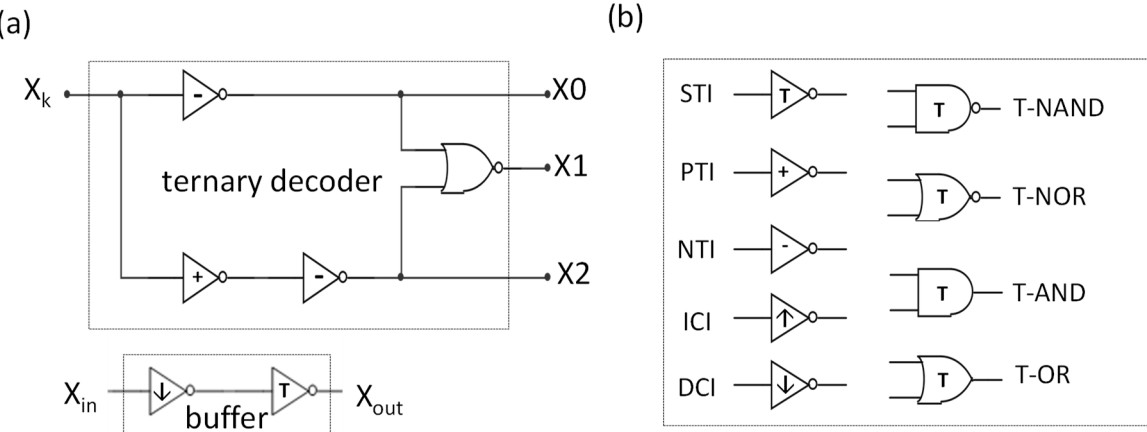

**Figure 4.** (**a**) Design of the ternary decoder and ternary buffer. (**b**) Symbols of all the ternary logic gates discussed in this work. Note: The unlabeled logic gates represent the normal binary logic gate.

### 3.2. Conventional Design of Ternary Adder

Adder is the most important arithmetic logical unit in a microprocessor, because all the mathematic operators and computations can be finally reduced into addition. Ternary logic decreases the requirement of components and interconnections by realizing more data transmission over an interconnection wire. Therefore, it is promising in high speed and high efficient data processing.

There are several ternary adders which have been designed in the published literatures, which can be generally categorized into two types: Capacitive-based and transistors-based circuits. The capacitive-based design benefits low complexity and low device count [15], however it exhibits high variation sensitivity and high area-cost because of the existing of capacitor. Most of the previous works focused on the transistor-based circuits where various logic gates are utilized to process the input and output voltage information [11–14]. In a conventional wisdom design of a transistor-based ternary ripple-carry adder, the two input numbers are summed together bit by bit, from the lowest address to the highest address. Though ternary intrinsically contains larger logic states densities, the existing designs of the ternary adder are not cost-efficient when compared to the binary adder. Figure 5 presents the conventional ternary half-adder design [11]. The ternary input A and B signals are firstly decoded by the ternary decoder (the two blue dashed box regions). Then they will be transferred to the next logic unit (red dashed box region) for computing. The output equations are given by Sum = (A2·B0 + A1·B1 + A0·B2) + 1·(A1·B0 + A0·B1 + A2·B2) and Carry = 1·(A2·B1 + A2·B2 + A1·B2). Note that the output of the decoder has only two logic values, i.e., "2" and "0", corresponding to logic "1" and "0" in binary logic. Therefore, binary AND/OR gates can be used here (Note: The logic states are still named as logic "2" and logic "0" in those binary logic gates). Finally the calculated binary results will be re-transferred into a ternary signal as the output. Two ternary buffers are needed here, one for the sum1 result (A1·B0 + A0·B1 + A2·B2) and the other one for the carry (A2·B1 + A2·B2 + A1·B2), because the binary signal of sum1 and carry can only be "0" or "1", and will never be "2".

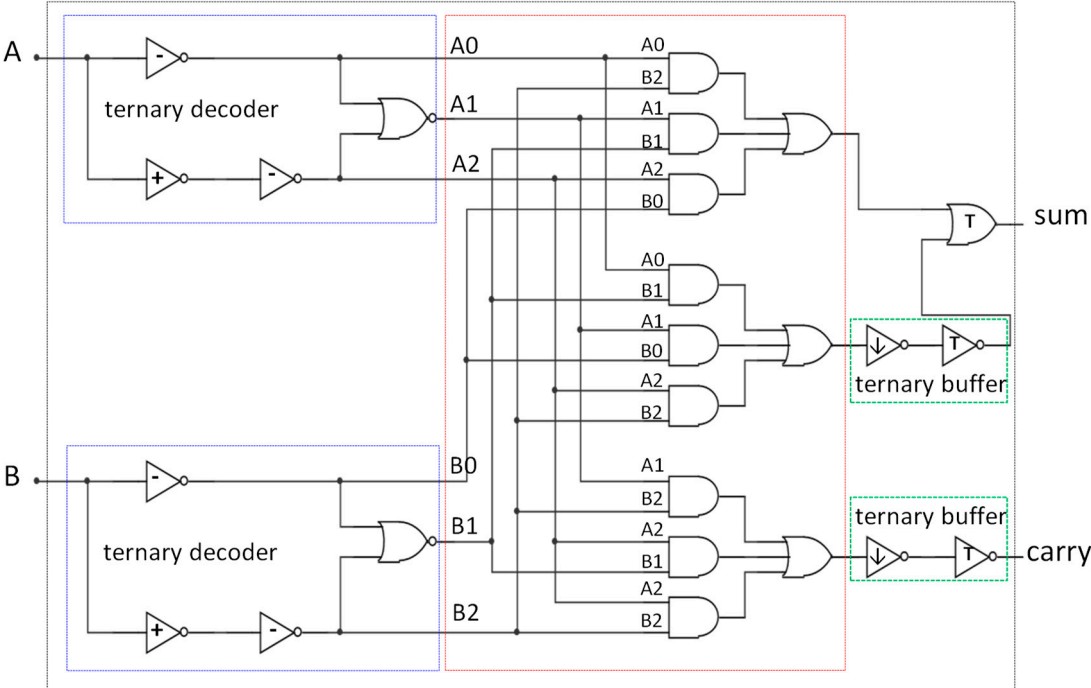

**Figure 5.** Conventional design of ternary half-adder. The ternary input (**A**, **B**) signals are firstly decoded by the ternary decoder (the two blue dashed box regions). Then the signals are transferred to the logic computation unit (red dashed box region). Finally the calculated results are re-transferred to be a ternary signal (green dashed box region) as output. Note: The unlabeled logic gates represent the normal binary logic gate.

Such ternary adder design has largely increased the circuitry complexity because it avoided the advantages of ternary logic. In fact, the main computation part is indeed made of binary logic gates. It has also largely increased the required transistors. For instance, a silicon-based, ternary half-adder requires 114 transistors and a 1-trit full-adder requires more than 200 transistors [5]. Meanwhile, a classical silicon CMOS binary only requires 28 transistors for the half-adder, and 62 transistors for the 1-bit full-adder. To simplify the circuit design and reduce the required transistors, we develop a mixed ternary-binary computation system where ternary cycling gates and the improved ternary adder algorithm are used.

### 3.3. Our Optimized Design of Ternary Adder

Firstly, for the ternary cycling gates, as we have discussed before, they can operate the increment function (output = input + 1) or decrement function (output = input − 1) which are especially suitable for the ternary adder. However, it requires more than 20 extra transistors to build such a cycling gate in the current existing ternary technology [11], therefore it is not cost-efficient to be applied into the ternary adder. Whereas, in our design, we can generate the ternary decrement cycling logic function by using only two transistors, as discussed in Figure 2e. Hence the DCI can be used in the half-adder to replace the original required AND gates, as shown in Figure 6a. The red dashed box presents our ternary computation method. If the input A = 0, the sum of A + B is B. If the input A = 1, the sum of A + B is B + 1 which needs one ICI gate (or two DCI gates). If the input A = 2, the sum of A + B is B + 2 = B − 1 which needs only one DCI gate. The calculated results are controlled by the transmission gate (1 enhancement-mode N-type transistor) and summarized in a ternary OR gate.

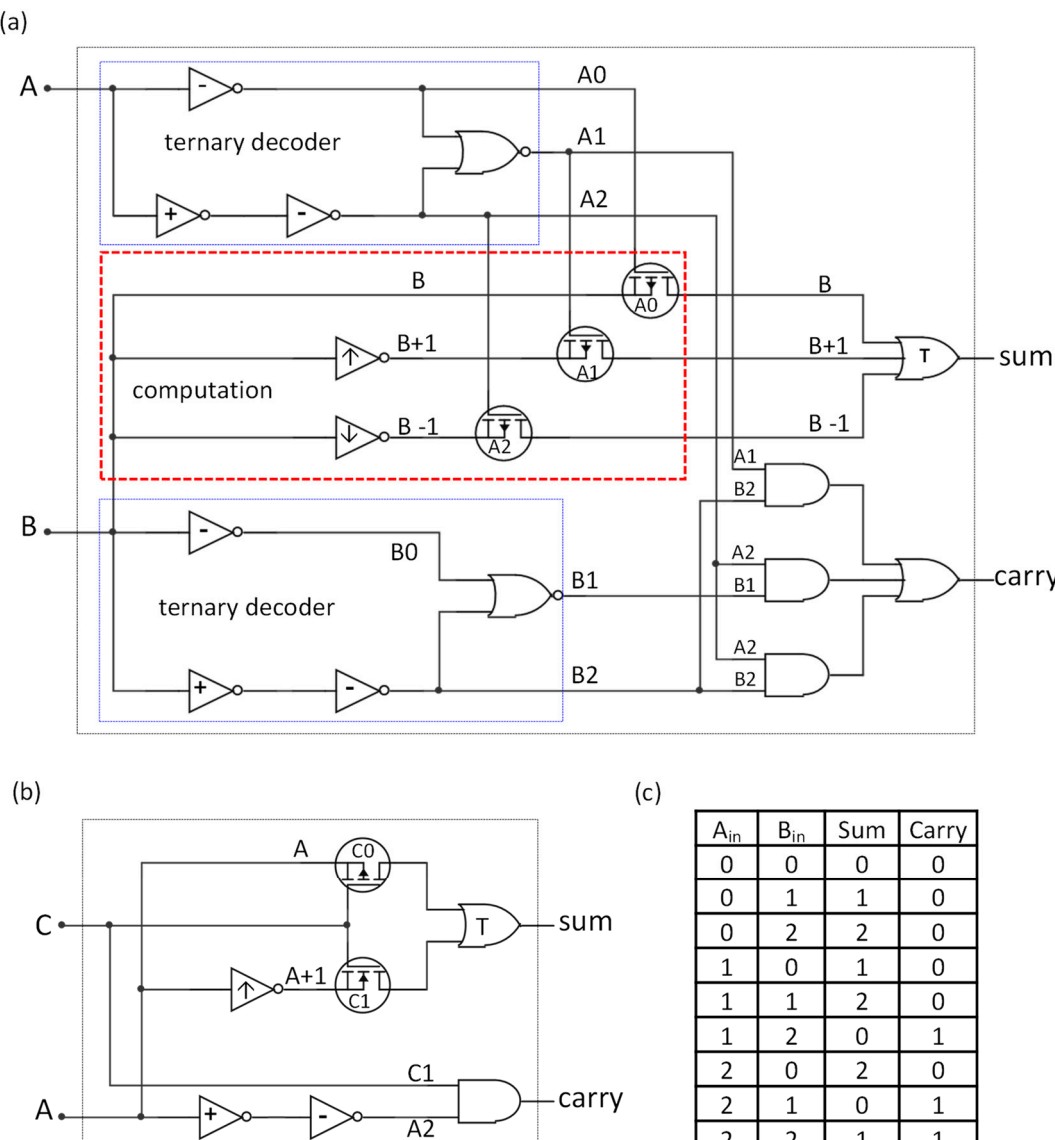

**Figure 6.** (**a**) Ternary half-adder Design 1, with the cycling DCI gate as the computation part. (**b**) Ternary half-adder Design 2, with the improved ternary adder algorithm, that the carry of the ternary adder can only be logic "0" or logic "1". (**c**) Truth tables of the half adder.

Next, note that the logic state of carry in the ternary adder (Figure 6c) can only be "0" or "1". For example, the carry of 0 + 0, 0 + 1, 0 + 2 and 1 + 1 is "0", and only the carry of 1 + 2 or 2 + 2 is "1". Therefore, it is not necessary to use the full ternary decoder. The improved ternary half adder is presented in Figure 6b. When C is "0", the sum is A; when C is "1", the sum is A + 1. While the carry is always C1 + A2.

Finally, the two as-designed half-adders can be combined into a 1-trit ternary full-adder, as shown in Figure 7a. The operating principle can be explained as follows: Step 1, the input A and B are summed up in the half-adder 1 (blue dashed box) and they then generate the result of AB_sum and AB_carry. Step 2, the AB_sum and the input carry of $C_{in}$ are summed up in the half-adder 2 (red dashed box). The generated result of SUM will be the final summation of A, B and $C_{in}$. Step 3, the two generated carry trits, AB_carry from the half-adder 1 and ABC_carry from the half-adder 2, will be sent into an OR gate, and will generate the final CARRY, which will be a new $C_{in}$ to the next series of full-adder.

In Figure 7b, we present the design block diagram of the 19-trit ripple-carry adder, in which A18A17 ... A1A0 is a ternary number of A input, B18B17 ... B1B0 is a ternary number of B input,

S18S17 . . . S1S0 is the output summation result, C18C17 . . . C1 is the Carry, C0 is always 0 and $C_{out}$ is the overflow bit.

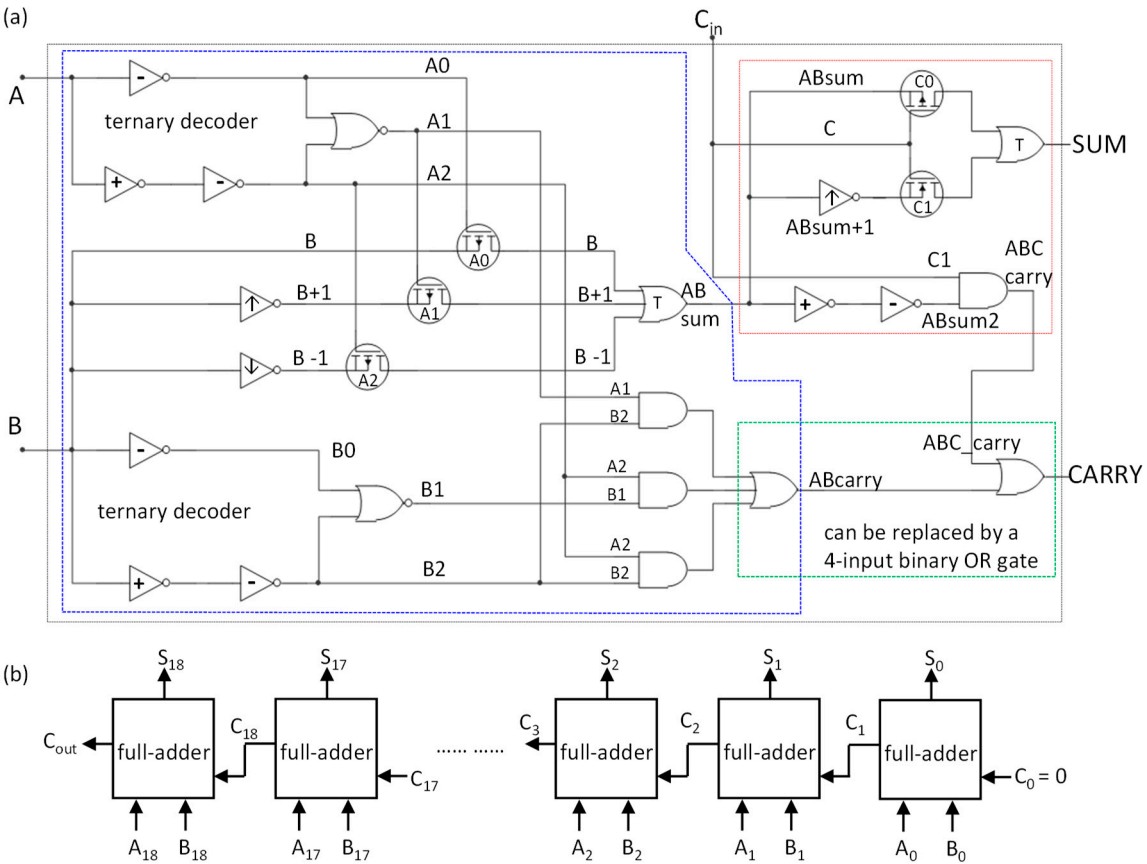

**Figure 7.** (**a**) Design of 1-trit full-adder. (**b**) Schematic diagram of the 19-trit ternary adder. A18A17 . . . A1A0 and B18B17 . . . B1B0 is ternary number of A and B, respectively. S18S17 . . . S1S0 is the output summation result, C18C17 . . . C1 is the Carry and the first input carry of C0 is 0.

By taking advantage of the ternary cycling inverter and the improved ternary adder algorithm, we present a mixed ternary-binary system where ternary gates are used for the calculation of SUM, and binary gates are used for the CARRY. Such design has largely decreased the required number of transistors, in which there are only 72, and 25 transistors are involved in half-adder 1 and half-adder 2, respectively. The total number of transistors in a full-ternary-adder is 99, which is much lower than that of the existing ternary circuits. The detailed comparison result can be seen in Table 1. More importantly, compared to the classical CMOS binary adder, our design shows a 7% reduction in the number of required transistors at the same data throughput. Another important parameter is the computation step, which has a direct impact on the computation speed. For the CMOS binary adder, the propagation delay of the logical gates in a 1-bit full binary adder is 10 (note that one AND gate or OR gate requires two times of gate delay). Therefore, the total computation steps for a 30-bit binary adder is 10*30 = 300. While that of our ternary ripple-carry adder is only 12*19 = 228. In consideration of the data throughput, the normalized required computation steps in our design show a 30% reduction compared to the binary design.

**Table 1.** Comparison table of different ternary circuit designs. (Note: The number in the table should be the less, the better.).

| | CMOS Binary | CMOS Ternary | CNT Ternary Circuits | | | 2D Ternary |
|---|---|---|---|---|---|---|
| | Circuits | ref.5 | ref.11 | ref.12 | ref.15 | (This Work) |
| Circuit structures | XOR, AND, OR gates | Decoder Binary logic gates | Decoder Binary logic gates | Transistor networks | 1.capacitive circuits 2.decoder 3.transistor networks | 1. Decoder 2. Ternary logic gates 3. Binary logic gates |
| Description | **Classical design** | **Classical design** | **Classical design** Similar to CMOS ternary | **Optimized design** with networks. **Reduced delay** | **Optimized design** with capacitors. **High speed and energy-efficient** | **Optimized design** with ternary gate. **Reduced circuits area** |
| number of FETs in half adder | 28 | 114 | 118 | - | - | design1: 72 design2: 25 |
| number of FETs in full adder | 62 | 234 | 242 | 114 | 2 capacitors2 decoders44 transistors | 99 |
| computation steps per bit/trit | 10 | 22 | 22 | - | - | 12 |
| ~1G data size — serial levels | 30 | 19 | 19 | 19 | 19 | 19 |
| ~1G data size — number of FETs | $62 \times 30 = 1860$ | $234 \times 19 = 4446$ | $242 \times 19 = 4598$ | $114 \times 19 = 2166$ | - | $99 \times 19 = 1881$ |
| ~1G data size — computation steps | 300 | 418 | 418 | - | - | 228 |
| Normalized required computation steps | 100% | 129% | 129% | - | - | 70% |
| Normalized required transistors | 100% | 221% | 228% | 107% | - | 93% |

## 4. Conclusions

In summary, we perform a systematical study on the 2D-materials-based ternary logic from individual ternary logic gates to large scale integrated circuits. We design and fabricate various ternary logic gates with different logic functions, which show good ternary performance with simplified circuital structure compared to traditional silicon ternary and CNT ternary. Then by developing the ternary cycling gate and the improved ternary adder algorithm, we propose a 19-trit ternary adder design with great circuitry simplicity. The design shows about a 50% reduction in the required number of transistors compared to the existing CNT ternary and silicon ternary, and it also shows itself competitive to the classic CMOS binary design with potential reduction in the number of required transistors and computation steps at the same data throughput. This work shows the potential for ternary logic in future integrated circuits application with higher data density, smaller chip area, faster computation speed and less latency.

**Author Contributions:** Conceptualization, B.T.; methodology, M.H.; validation, M.H., X.W. and G.Z.; data curation, M.H.; writing—original draft preparation, M.H.; writing—review and editing, X.W., M.H., G.Z., P.C. and B.T.; supervision, B.T. and P.C.

**Funding:** This work was supported by Ministry of Education, Singapore (Grant No.: MOE2015-T2-2-043, MOE2017-T1-002-200).

**Conflicts of Interest:** The authors declare no conflict of interest.

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
