# Peer review of "Design and Implementation of Ternary Logic Integrated Circuits by Using Novel Two-Dimensional Materials"

_applsci, doi:10.3390/app9204212_

Round 1

Reviewer 1 Report

The authors work is novel and commendable. Shed light in the possibility of developing ternary systems with 2D materials. 

The methodology is missing, and the authors should provide a method section where they describe how they arrived at their results so that other researchers can replicate and/or apply it to other systems. 

Reviewer 2 Report

The authors present how the novel 2D materials MoS2 and black phosphorous (BP) can be used to design four different ternary logic gates. They are then able to demonstrate through circuit design how various different ternary logic circuits could be designed. Finally and most impressively they are able to demonstrate that these circuits would be superior to previous ternary circuits based on Si or carbon nanotubes (CNT).

However despite the significance of these results I feel there is one major flaw with the manuscript as things stand that currently prevents it being suitable for publication. The authors provide absolutely no information as to how they fabricate the ternary logic gates for which they presents the Vout - Vin curves? The authors need to detail how they obtain the 2D materials they use (and show characterisation of them to verify that they are pure and not oxidised or in other was damaged etc.) and then detail how they fabricate the logic gates incorporating the 2D materials including diagrams, pictures instructions etc. If and when the authors provide this information then this manuscript should then be very worthy of publication.

In addition the following smaller issues need addressing before the manuscript could be published.

1) Whilst the quality of the writing is generally very good for which the authors should be commended, a thorough proof read would still be welcome. Particularly lots of acronyms are used without first being defined and this should be rectified.

2) In the introduction no mention is made of previous literature on ternary logic based on 2D materials (and there is some literature out there). This should be appropriately referenced and the merits of this work versus the previous work should be highlighted.

3) As stated above a whole section on 2D material synthesis/chracterisation and device fabrication needs to be added.

4) Whilst the Vout - Vin curves presented in Figure 1 make clear that the four different ternary logic gates were made and operate correctly, it is not clear to me whether the authors made any of the ternary logic circuits they describe in the subsequent sections? Or are they just saying that based on the ternary logic gates they have been able to make then they would be able to make those circuits? This needs to be made clearer.

Reviewer 3 Report

I commend the authors for the interesting work.

There is recent research that suggests that Carbon Nanotubes are becoming more viable and full binary processors can be designed using them. Check the Stanford research in that regard. Also, the recently published works in that regard. The authors show that their design shows 50% transistors than modern ternary implementations. I would be interested in seeing speed, energy and EDP numbers on top of the raw transistor count. 

Ref 36 is implementing a ternary multiplier using CNTFETs at 32nm technology node only. There is more recent work in the area that would compare better to this work.

Ref 37 copied below is a 1978 patent. Don't you think this is kind of outdated? Is this the state of the art in the subject? 

37. Moufah H T. Ternary logic circuits with CMOS integrated circuits: U.S. Patent 4,107,549[P]. 1978, 8-15. 

It would be good to show comparisons against recent relevant CNT implementations. 

For example, 

https://www.sciencedirect.com/science/article/pii/S0045790618321955

https://arxiv.org/abs/1806.07570

you can definitely find other similar or relevant comparisons. You can compare the reported results in these and similar papers. I would like to see comparisons against several CNT implementations, not just one.
